# Hybrid Epithelial/Mesenchymal State in Cancer Metastasis: Clinical Significance and Regulatory Mechanisms

**DOI:** 10.3390/cells9030623

**Published:** 2020-03-04

**Authors:** Tsai-Tsen Liao, Muh-Hwa Yang

**Affiliations:** 1Graduate Institute of Medical Sciences, College of Medicine, Taipei Medical University, Taipei 110, Taiwan; liaotsaitsen@tmu.edu.tw; 2Cell Physiology and Molecular Image Research Center, Wan Fang Hospital, Taipei Medical University, Taipei 110, Taiwan; 3Institute of Clinical Medicine, National Yang-Ming University, Taipei 112, Taiwan; 4Cancer Progression Research Center, National Yang-Ming University, Taipei 112, Taiwan; 5Division of Medical Oncology, Department of Oncology, Taipei Veterans General Hospital, Taipei 112, Taiwan

**Keywords:** cancer, epithelial-mesenchymal transition, hybrid, metastasis

## Abstract

Epithelial-mesenchymal transition (EMT) has been well recognized for its essential role in cancer progression as well as normal tissue development. In cancer cells, activation of EMT permits the cells to acquire migratory and invasive abilities and stem-like properties. However, simple categorization of cancer cells into epithelial and mesenchymal phenotypes misleads the understanding of the complicated metastatic process, and contradictory results from different studies also indicate the limitation of application of EMT theory in cancer metastasis. Nowadays, growing evidence suggests the existence of an intermediate status between epithelial and mesenchymal phenotypes, i.e., the “hybrid epithelial-mesenchymal (hybrid E/M)” state, provides a possible explanation for those conflicting results. Appearance of hybrid E/M phenotype offers a more plastic status for cancer cells to adapt the stressful environment for proceeding metastasis. In this article, we review the biological importance of the dynamic changes between the epithelial and the mesenchymal states. The regulatory mechanisms encompassing the translational, post-translational, and epigenetic control for this complex and plastic status are also discussed.

## 1. Introduction 

During embryonic development, epithelial cells lose their polarity and undergo multiple biochemical changes to convert into a mesenchymal phenotype for acquiring the migratory capacity for morphogenesis. Elizabeth Hay first described the process as “epithelial-mesenchymal transformation”, which is later referred as EMT [1], through observation of the chick primitive streak formation model in the 1980s [2,3]. Based on the biological setting of EMT, the EMT was further categorized into three different subtypes: Type 1, EMT associated with development, including implantation, gastrulation, and neural crest formation; Type 2, EMT associated with tissue regeneration and organ fibrosis; and Type 3, EMT associated with cancer progression and metastasis [4]. Regarding the cancer cells undergoing EMT, the intercellular junctions are disrupted, the cellular polarities are lost, the cytoskeletons are reorganized, and the cell motilities are increased. Beyond allowing epithelial cancer cells to acquire migratory and invasive characteristics, EMT also overcomes senescence, apoptosis, and anoikis, which are essential properties for cancer dissemination and metastasis [5]. Interestingly, the original EMT study noticed that embryonic mesenchymal cells re-exhibit the epithelial phenotype [3]. The reverse process of EMT is later referred as mesenchymal-epithelial transition (MET) and recognized as an essential process for colonization during metastasis [6]. The clinical fact that most metastatic tumors exhibit an epithelial-phenotype histology rather than a mesenchymal phenotype also supports the occurrence of reversion of EMT during metastatic colonization [7].

In the past decades, most experimental models define EMT by examining the expression of epithelial and mesenchymal makers, EMT transcriptional factors (EMT–TFs), and morphological changes. Canonical epithelial markers include E-cadherin, epithelial cell adhesion molecule (EpCAM), cytokeratin, and occludin; mesenchymal markers contain N-cadherin and vimentin [8]. However, this binary system is not satisfied to explain the real phenomena in the clinical setting, and increasing evidence discloses the concept of EMT as a “spectrum” rather than a binary status, which highlights the importance of the status of intermediate, partial, or “hybrid epithelial/mesenchymal (hybrid E/M)”. The plasticity of the hybrid status allows cancer cells to adopt the environmental stress during malignant progression. Furthermore, the intermediate-state concept provides clear connections between the epithelial plasticity and collective migration, cancer stemness, metastatic ability, and resistance to therapy [3,9]. In recent years, advances in technology made scientists possible to interrogate the dynamic changes of the hybrid E/M phenotype. In this paper, we will review the latest progress of hybrid E/M in cancer metastasis.

## 2. Evidence of Hybrid E/M in Cancers

The hybrid E/M phenotype or partial EMT has been observed in clinical cancer patients. An important example is the existence of hybrid phenotype in circulating tumor cells (CTCs) [10]. The number of CTCs has been regarded as a biomarker for indicating patient prognosis and treatment efficacy. The current method for CTCs isolation relies exclusively on the detection of the CTCs expressing epithelial-lineage markers (e.g., EpCAM and cytokeratins) in the peripheral blood [11,12,13]. However, in patients with advanced cancers, a high frequency of CTCs expressing both epithelial and mesenchymal markers was noted [14,15]. Interestingly, these CTCs show features of collective migration, in which cells migrate with retained cell–cell contacts and disseminate as tumor cell clusters [16,17]. Moreover, the EMT spectrum are demonstrated in the cellular clusters: the leading cells present hybrid E/M phenotype with both increased mesenchymal phenotype and higher actin-mediated mobility; whereas the rest cells in the center of the clusters maintain their polarity and intercellular junctions and migrate alone with the traction forces generated by leader cells [18]. In contrast to collective migration, single-cell migration usually requires a full EMT phenotype with the loss of cell adhesion and apicobasal polarity, gain of front–back polarity, and an increase of individual cell motility [18]. Nevertheless, the disseminated individual cancer cells are barely detected in clinical tumor samples; on the contrary, the clusters of cancer cells are frequently observed in the invasive front of tumors [19,20].

In addition to CTCs, the hybrid E/M phenotype was also observed in primary human cancers including prostate cancer, breast cancer, and lung cancer [21,22,23,24]. However, the EMT statue of human cancers was mostly characterized by the bulk tumor transcriptomes, and recent evidence indicates that the mesenchymal gene expression of the hybrid E/M status may likely be originated from the stromal cells in and around the tumors [25]. Single-cell RNA sequencing was used to distinguish the mesenchymal transcriptomes between stromal cells and EMT tumor cells. The analysis of head and neck squamous cell carcinoma (HNSCC) indicated that the hybrid E/M subpopulation localizes at the leading edge of primary tumors [26]. The hybrid E/M subtype is an independent predictor of nodal metastasis, histological grade, and adverse pathologic features in HNSCC [26]. The hybrid E/M state was also observed in the patient-derived xenotransplantation (PDX) model of human lung cancer, breast cancer, and colorectal cancer [27,28]. In the PDX model, the human stroma is replaced by mouse cells after serial passages that allow us to distinguish the changes of epithelial markers of human cancer cells from mouse stroma [28,29,30]. The tumors from the spontaneous mouse prostate cancer model also showed the existence of three sub-populations: epithelial (EpCAM^+^Vim^–^), hybrid E/M (EpCAM^+^Vim^+^), and mesenchymal (EpCAM–Vim^+^). Interestingly, the isolated epithelial and mesenchymal cells mainly remain in their initial cell state while culturing, whereas the majority of hybrid E/M tumor cells transit into fully epithelial or mesenchymal state as early as 24 h after plating, demonstrating the plastic and transitory state of hybrid E/M and the importance of microenvironment for maintaining this phenotype [31]. 

The hybrid E/M state was also noted in breast, ovarian, lung, colorectal cancer, prostate cancer, and renal cancer cell lines in vitro [9,32,33,34,35,36,37]. A large-scale EMT scoring matrix indicated that the origin of cancer cell lines could be considered as a good indicator of the EMT status in most cases; however, whether these phenotypic traits are inherent or acquired is still unclear [38]. Furthermore, the cancer cell lines with different EMT spectrum are derived from cancers with heterogeneous clinical stages, mutation profiles, and epigenetics. All these factors bring great diversities even in the same cancer type, and it is essential to consider these factors while applying cancer cell lines to study hybrid E/M in vitro.

## 3. The Regulation of Hybrid E/M in Cancer Cells

During cancer progression, EMT is triggered by transcriptional and epigenetic control through coordinated regulation by the EMT transcription factors (EMT–TFs), microRNAs, and epigenetic modulators [3]. Forced expression of EMT–TFs such as Snail, Twist1, ZEB1, and ZEB2 promote EMT and increase their ability to give rise to secondary tumors, where miRNAs are shown regulate EMT–TFs through selectively repress the target mRNAs via cleavage degradation or translational repression [39,40]. In the following section, we will review the regulatory mechanisms of the binary epithelia/mesenchymal and intermediate states. Understanding the regulation of binary/intermediate EMT will be helpful for modeling the mechanisms for hybrid E/M in cancer cells.

### 3.1. Conventional EMT Inducers

The key EMT–TFs serve as the master regulators to repress the epithelial genes and induce mesenchymal genes. For example, one of the most famous EMT–TFs Snail functions as the transcriptional repressor for epithelial genes by binding to the E-boxes at the promoters of the genes encoding intercellular junction proteins and recruiting histone modifiers to repress their transcription [41,42,43,44,45,46,47]. Snail also functions as an activator for mesenchymal genes and contributes to the mesenchymal phenotype [48,49,50]. The similar mechanism is also noted in another major EMT–TF ZEB1 which is originally known to bind to E-boxes for repressing E-cadherin expression [51,52]. ZEB1 also acts as an activator by interacting with Smads and the transcriptional co-activator p300 to induce EMT [53,54,55]. Several key molecules are noted to control EMT through regulation of the expression of EMT–TFs. For example, ovo like zinc finger 2 (OVOL2) restricts EMT by directly repressing ZEB1 and induces MET [56,57,58,59]. MiR-200 family miRNAs repress the expression of ZEB1 and ZEB2, thereby maintaining cancer cells in the epithelial phenotype [60,61,62]. ZEB1 and miR-200 family members exist a double-negative feedback loop to control the balance between epithelial and mesenchymal states [63,64]. Despite the regulations of epithelial/mesenchymal gene expression program by EMT–TFs have been demonstrated extensively, there have been debates for the essential role of “canonical EMT” in cancer metastasis. For example, the EMT–TFs Snail and Twist1 promote metastasis in various types of cancers including breast cancer, lung cancer, and HNSCC [19,65,66]; however, an independent study revealed that Snail or Twist1 is dispensable for metastasis in pancreatic cancer [67]. These findings implicate the possibility about the favorable role of the coexistence of epithelial and mesenchymal phenotypes in metastasis.

Post-translational modifications are crucial events for determining the activity of EMT–TFs. For instance, GSK-3β was shown to phosphorylate Snail at two consensus motifs for ubiquitin-mediated degradation and nuclear export [65]. Suppression of GSK-3β-mediated phosphorylation of Snail induces EMT [68,69,70]. Furthermore, the phosphorylation of Snail also controls the subcellular localization via other kinases including PDK1, LATS2, and PAK1 in cancer cells [71,72,73]. In addition to serine/threonine phosphorylation, lysine acetylation is another critical event for determining Snail activity. We previously demonstrated that acetylation of Snail by CBP/p300 at lysine 146 and lysine 187 switches its function from a repressor to an activator via recruiting different histone modifiers [74]. These findings implicate that transcriptional repression of junctional proteins alone is not capable of completing metastatic process, and expression of both epithelial and mesenchymal markers may be important for cancer progression. Additional evidence supports that loss of E-cadherin did not induce broad mesenchymal gene expression and cells may retain their epithelial identity [75]. Recently, E-cadherin has been shown to function as a survival factor in invasive breast cancers during the detachment, systemic dissemination, and seeding phases of metastasis by limiting reactive oxygen-mediated apoptosis [76]. Re-expression of E-cadherin occurs early rather than just for colonization, the last step of metatsasis [76], which also indicates the essential role of hybrid E/M status in cancer metastasis. Re-localization of epithelial proteins leads to a partial EMT phenotype, which promote formation of CTC clusters and collective migration [77]. We recently revealed that acetylated Snail prompts collective migration, a hallmark of the hybrid E/M phenotype, in 2.5D culture and formation of circulating tumor clusters in HNSCC patients [78]. These results indicate that the EMT–TFs prompt cancer cells to maintain in a phenotype located in the medium of epithelial–mesenchymal spectrum rather than completed EMT, which highlights the context dependency of the EMT–TFs in response to different environments/experimental models [79]. 

Other studies also showed that cancer cells are frequently found to be in different EMT spectrums, which are in a delicate equilibrium and reflect the propensity of cancer to disseminate and become refractory to therapy. For example, the coexistence of the prostate cancer cells with heterogeneous epithelial and mesenchymal subpopulations enhances the local invasiveness of the epithelial subpopulation, thus contributing to the overall metastatic potential of the tumor [80]. Another study showed an increased hybrid E/M population with co-expression of E-cadherin and vimentin, collective cell migration, and stem-like properties in the erlotinib-resistant subline of HCC827 lung cancer cells [81]. In breast cancer, hybrid E/M cells are associated with adaptive resistance to tamoxifen, trastuzumab, and taxanes [82,83,84,85]. Collectively, EMT–TFs may induce hybrid E/M state instead of completed EMT, and the hybrid state is favorable for metastasis and drug resistance.

### 3.2. Transcriptional Regulation of Hybrid E/M

In contrast to the well-characterized factors decide the binary fates, the hybrid E/M phenotype harbors a greater complexity and diversity [3]. Most studies applied the canonical epithelial and mesenchymal makers combined with different factors to define hybrid E/M. Several studies used the mathematic model in the well-characterized factors to describe the dynamics of hybrid E/M, such as the miR-34/Snail and the miR-200/ZEB mutually inhibiting loop as a ternary switch between epithelial, mesenchymal, and hybrid E/M phenotypes [86,87,88]. The mathematic model was also used to seek the stabilizing factors for maintaining hybrid E/M phenotype. The identified factors include NRF2, Numb, OVOL2, and GRHL2. These factors predict poor outcome of patients and the knockdown of them drive a complete EMT [87,89]. Decreased expression of OVOL2 or GRHL2 impairs collective cell migration [87]. Numb stabilizes hybrid E/M phenotype through Notch signaling pathway [89,90]. Notch1 transactivates Notch3 and they act cooperatively to drive epithelial differentiation in oral cancer. A distinct effect of Notch1 was shown in the presence of TGF–β: Notch1 represses Notch3 and induces ZEB1 expression, which limits full EMT for permission of the hybrid E/M status [91]. Altogether, the computational approaches provide systematic tools to identify the factors that control the different states spanning the EM spectrum; nevertheless, experimental validations are mandatory for these predictions.

The hybrid E/M status can be further divided into multiple states based on the combination of different markers. In a genetically-modified skin squamous cell carcinoma model that tumor cells undergo EMT spontaneously, the tumors contain Epcam^+^ epithelial and Epcam^–^ mesenchymal populations. Hybrid E/M status was divided into early and late hybrid E/M states according to the expression patterns of the surface markers CD106, CD61, and CD51 [28]. Several studies also reported different markers for distinguishing early/late hybrid EM state. The reported markers for dissecting hybrid E/M is summarized in Figure 1. These studies provide valuable information for understanding the characteristics of tumor cells such as stemness and metastatic capability with different epithelial/mesenchymal states [28,34,92].

### 3.3. Epigenetic Regulation of Hybrid E/M

During different steps of metastasis, cancer cells undergo dynamic and reversible transitions between epithelial and mesenchymal phenotypes to endure the environmental stress and increase the chances of successful metastasis. Epigenetic regulation of the transcription of epithelial/mesenchymal genes enables the dynamic changes and plasticity between epithelial and mesenchymal states. Extensive studies support the indispensable role of epigenetic regulation in the induction of EMT [93] as well as generation of cancer stem cells (CSCs) [94]. Specific chromatin modifiers including histone deacetylases and polycomb group proteins participate in EMT, resulting in a transcriptome drift to the mesenchymal-like CSCs [65,75,95,96]. Repression of epithelial genes is regulated by the enrichment of H3K27me3 to form a bivalent modification with H3K4me3 and create a highly plastic and reversible state [97,98]. Furthermore, losing the active code H3K4me3 facilitates the subsequent formation of the heterochromatic modification H3K9me3, which is more stable and enhance the recruitment of DNA methyltransferases (DNMTs). DNA methylation on the epithelial gene promoters creates the highly stable methylated CpG dinucleotides that can be perpetuated over many cell generations [98]. For example, Snail recruits polycomb repressive complex 2 (PRC2) to repress *CDH1* expression through increasing H3K27me3 on the promoter of *CDH1* [45]. In ovarian cancer cells, epithelial genes are more susceptible to epigenetic reprogramming by CpG methylation and histone H3 modifications [99].

In hybrid E/M state, the epigenetic landscape shows that the repressed promoters have high H3K27me3 and low H3K4me3, whereas the activated promoters harbor high H3K4me3 and high H3K27ac. A repressed enhancer region is characterized by H3K4me1 with/without H3K27me3, whereas an active enhancer is characterized by H3K4me1 with high H3K27ac [99]. GRHL2, which is recognized as the pioneer factor for regulation of the chromatin accessibility, inhibits the repressive activities of EMT–TFs and/or epigenetic repressors such as PRC2 complex, histone deacetylases (HDACs) and DNMTs at promoters and/or enhancers of epithelial genes [99,100,101]. GRHL2 was shown to be involved in the epigenetic control during the intermediate phases of EMT/MET [99]. The chromatin modifier HMGA2 is also noted to regulate the epithelial–mesenchymal plasticity and is significantly upregulated in hybrid E/M and mesenchymal state of the mouse prostate tumor cells [31].

A previous study applied assay for transposase-accessible chromatin using sequencing (ATAC-seq) with transcriptional profiling to define transcriptional and chromatin landscapes in different epithelial/mesenchymal states. It found that ΔNP63 promotes the entrance of hybrid E/M in squamous cell carcinoma [28]. Meanwhile, AP1, Ets, Tead, and Runx motifs are enriched at transition states, suggesting the preserved transcription factors are required to induce chromatin remodeling of the intermediate state of EMT [28,102]. However, the current understanding of the epigenetic regulation in hybrid E/M is relatively limited, and single-cell level studies are mandatory to provide a more comprehensive viewpoint for hybrid E/M.

## 4. Hybrid E/M and Cancer Stemness

CSCs are a subpopulation of cancer cells with the abilities of self-renewal, tumor initiation, metastasis, and resistance to chemotherapy. CSCs also provide the phenotypic heterogeneity of tumor cells [103,104]. The stem-like properties of cancer cells are generally validated with the expression cancer stem cell markers, ability of tumorspheres formation, and in vivo tumor initiation. EMT process permits cancer cells to acquire stem cell properties for metastasis and dissemination. For example, ectopic expression of Snail/Twist1 in cancer cells results in the changes of the surface marker to a stem-like phenotype (CD44^high^/CD 24^low^) and enhances the mammosphere-forming ability [94]. We previous showed that Twist1 acts collaboratively with the chromatin modifier Bmi1 to suppress the expression of let-7, a microRNA expressed during stem cell differentiation, leading to increased stemness in HNSCC [105,106]. However, a study reveals that in human breast cancer cells, knockdown of paired-related homeobox transcription factor 1 (Prrx1), a recently identified EMT inducer, increased mammosphere formation, self-renewal capacity, and CD44^high^/CD24^low^ CSCs [107]. The contradictory findings in different studies implicate that the hybrid E/M rather than the completed epithelial or mesenchymal state is more likely to acquire stemness. For instance, transient expression of Twist1 induces long-term invasiveness and colonization capability by promoting the coexistence of the epithelial and mesenchymal cellular feature [108]. The hybrid E/M populations also shows a five times increase in tumor propagation compared to epithelial tumor cells [28]. These results suggest that hybrid E/M state is more flexible and harbors a higher potential to acquire stem-like properties [109]. Since the cellular plasticity is highly associated with stemness among different epithelial/mesenchymal states, some studies also used the stemness markers as the determinant for subgrouping hybrid E/M. In breast cancer, ZEB1 represses the expression of the epithelial transcription factor TAp63α (tumor protein 63 isoform 1) and promotes ITGB4 (also known as CD104) expression, which allows the cells to present as tumor-initiating cells. The ITGB4^+^ CSCs manifest a hybrid E/M state. In triple-negative breast cancer patients, elevated ITGB4 expression was associated with a worse relapse-free survival [34]. CD104/CD44 has been used to define cancer cells in different epithelial/mesenchymal states: the epithelial (CD104^+^CD44^low^), hybrid E/M (CD104^+^CD44^hi^), and mesenchymal (CD104^−^CD44^hi^) subpopulations [110]. Meanwhile, they also observed the dynamic expression rather than a fixed level of EMT–TFs in cancer cells. Snail is up-regulated in the hybrid E/M cells and down-regulated in the mesenchymal cells, whereas high ZEB1 expression was observed in the mesenchymal cells [110].

## 5. Conclusions

Growing evidence uncovers the essential role of the hybrid E/M state in tumor progression. Tumor cells are in different EMT spectrum rather than completed epithelial or mesenchymal state. Hybrid E/M state is associated with increased cellular plasticity, collective migration, stem-like properties, and metastatic potential. However, the knowledge of hybrid E/M is relatively limited compared to the extensive studies for EMT. Understanding the biological importance and regulatory mechanisms of hybrid E/M will be helpful in the development of therapeutic strategies for targeting these highly plastic and dynamic populations.

## Figures and Tables

**Figure 1 cells-09-00623-f001:**
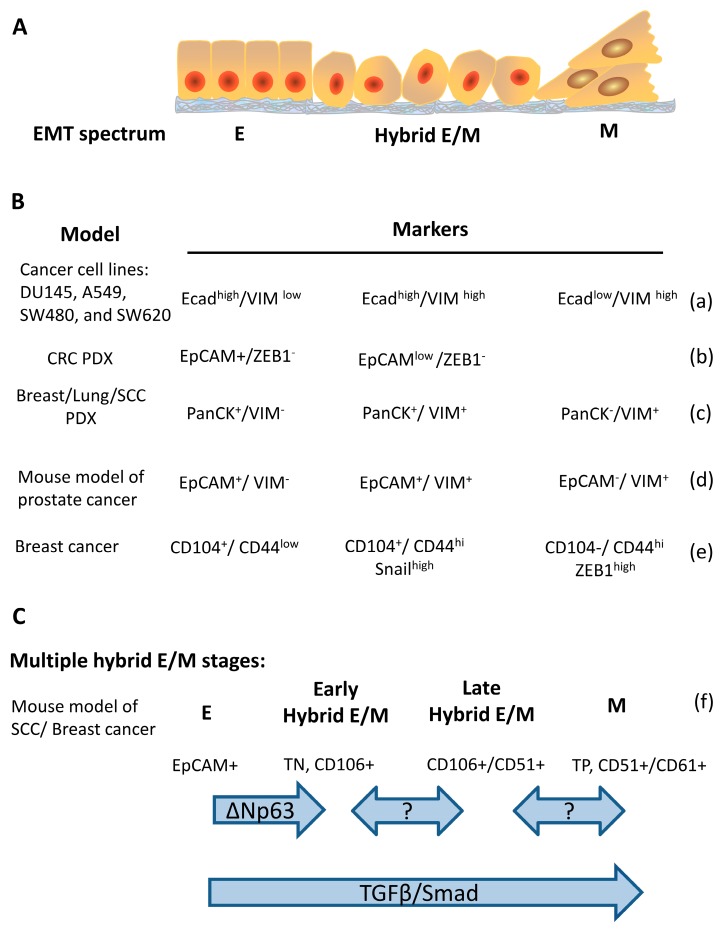
Markers and regulators of cancer cells along the epithelial–mesenchymal spectrum. (**A**) schematic representation of the cancer cells in epithelial, hybrid E/M, and mesenchymal states. (**B**) a table for summarizing the reported epithelial, hybrid E/M, and mesenchymal markers in different cancer cell lines/models. Ecad, E-cadherin; VIM, vimentin; PDX, patient-derived xenografts. (**C**) a schema for illustrating the markers along the EMT spectrum in which hybrid E/M is divided into early and late stages. TN, triple negative that means non-expression of CD51, CD61, and CD106; TP, triple positive that means co-expression of CD51, CD61, and CD106 (a) [9], (b) [27], (c) [28], (d) [31], (e) [38], (f) [28].

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
