# Peer review of "Hybrid Epithelial/Mesenchymal State in Cancer Metastasis: Clinical Significance and Regulatory Mechanisms"

_cells, 2020, doi:10.3390/cells9030623_

Round 1

Reviewer 1 Report

This review article from Liao and Yang is overall well written and laid out. It describes an interesting hybrid state between epithelial and mesenchymal states, which is of great interest for the field. I recommend some minor changes, to make the article more comprehensive and easier to read. 57 - Evidence of hybrid E/M in cancers It would be great to have a little more depth in this section. Multiple papers are mentioned and cited, but it would be nice to have more information regarding those studies in this review. E.g. how did those studies distinguish between epithelial, mesenchymal, and hybrid states? Which genes did they use to differentiate and why? 111 - 3.1 Describes EMT in cancer, and 3.2 hybrid E/M. It would be nice to have a chapter in between on MET, describing how it is induced/regulated in cancer, and what the known players in this process are, similar to how the authors described EMT in 3.1. This would provide a complete picture of the story. 209 - “losing the active code H3K4me3”. Should be “promoter mark” I recommend a native English speaker reads carefully over the manuscript and corrects minor grammar mistakes.

Author Response

Comment 1:

57 - Evidence of hybrid E/M in cancers It would be great to have a little more depth in this section. Multiple papers are mentioned and cited, but it would be nice to have more information regarding those studies in this review. E.g. how did those studies distinguish between epithelial, mesenchymal, and hybrid states? Which genes did they use to differentiate and why?

Authors’ reply:

Thank you for the comment. For detailing the difference between epithelial, mesenchymal, and hybrid states, we incorporate a paragraph as below.

In line 353-359 of the revised manuscript (line 96-102 without track changes):

Compared with the clearly defined epithelial and mesenchymal states, the hybrid E/M seems to lack universal markers to indicate the state. Most studies applied co-expression of epithelial and mesenchymal markers to indicate the hybrid E/M state [9,27,28,31]. Regarding the hybrid state-specific markers, a recent study shows that expression of CD 51, CD61, and CD106 indicates different hybrid E/M states [28]. Expression of CD104 was also noted in the hybrid E/M populations [38]. The transcriptional factor ΔNP63 engenders the entrance of an early hybrid E/M state [28]. The reported hybrid E/M markers/transcriptional factors are summarized in Figure 1.

In addition, we also mentioned this point in the Conclusion of the manuscript in line 1311-1316 of the revised manuscript (line 320-325 without track changes):

Specifically, previous studies mostly applied co-expression of epithelial and mesenchymal markers to define the hybrid E/M state. Nevertheless, co-expression of these markers may not be able to reflect the highly plastic characteristics of the hybrid E/M state, and defined markers for the hybrid E/M state will be important for both experimental and clinical characterization. Application of single-cell analysis techniques will be extremely helpful in solving this issue.

Comment 2:

111 - 3.1 Describes EMT in cancer, and 3.2 hybrid E/M. It would be nice to have a chapter in between on MET, describing how it is induced/regulated in cancer, and what the known players in this process are, similar to how the authors described EMT in 3.1. This would provide a complete picture of the story.

Authors’ reply:

Thank you for the suggestion. We added a chapter “MET for metastatic colonization “as 3.2 to address the MET part as the reviewer suggested, and shift the original 3.2 “hybrid E/M “to 3.3 so on.

Comment 3:

209 - “losing the active code H3K4me3”. Should be “promoter mark” I recommend a native English speaker reads carefully over the manuscript and corrects minor grammar mistakes. 

Authors’ reply:

We have revised the sentence according to the suggestion. We also send the manuscript for English editing. The certificate of the English editing is attached at the bottom of the letter.

Reference:

  1. George, J. T.; Jolly, M. K.; Xu, S.;  Somarelli, J. A.; Levine, H., Survival outcomes in cancer patients predicted by a partial EMT gene expression scoring metric. Cancer Res. 2017, 77 (22), 6415-6428.

  1. Mizukoshi, K.; Okazawa, Y.; Haeno, H.; Koyama, Y.; Sulidan, K.; Komiyama, H.; Saeki, H.; Ohtsuji, N.; Ito, Y.; Kojima, Y.; Goto, M.; Habu, S.; Hino, O.; Sakamoto, K.; Orimo, A., Metastatic seeding of human colon cancer cell clusters expressing the hybrid epithelial/mesenchymal state. Int J Cancer. 2019.

  1. Pastushenko, I.; Brisebarre, A.; Sifrim, A.; Fioramonti, M.; Revenco, T.; Boumahdi, S.; Van Keymeulen, A.; Brown, D.; Moers, V.; Lemaire, S.; De Clercq, S.; Minguijon, E.; Balsat, C.; Sokolow, Y.; Dubois, C.; De Cock, F.; Scozzaro, S.; Sopena, F.; Lanas, A.; D'Haene, N.; Salmon, I.; Marine, J. C.; Voet, T.;  Sotiropoulou, P. A.; Blanpain, C., Identification of the tumour transition states occurring during EMT. Nature 2018, 556 (7702), 463-468.

  1. Ruscetti, M.; Dadashian, E. L.; Guo, W.; Quach, B.; Mulholland, D. J.; Park, J. W.; Tran, L. M.; Kobayashi, N.; Bianchi-Frias, D.; Xing, Y.; Nelson, P. S.; Wu, H., HDAC inhibition impedes epithelial-mesenchymal plasticity and suppresses metastatic, castration-resistant prostate cancer. Oncogene 2016, 35 (29), 3781-95.
  1. George, J. T.; Jolly, M. K.; Xu, S.;  Somarelli, J. A.; Levine, H., Survival outcomes in cancer patients predicted by a partial EMT gene expression scoring metric. Cancer Res. 2017, 77 (22), 6415-6428.

  1. Mizukoshi, K.; Okazawa, Y.; Haeno, H.; Koyama, Y.; Sulidan, K.; Komiyama, H.; Saeki, H.; Ohtsuji, N.; Ito, Y.; Kojima, Y.; Goto, M.; Habu, S.; Hino, O.; Sakamoto, K.; Orimo, A., Metastatic seeding of human colon cancer cell clusters expressing the hybrid epithelial/mesenchymal state. Int J Cancer. 2019.

  1. Pastushenko, I.; Brisebarre, A.; Sifrim, A.; Fioramonti, M.; Revenco, T.; Boumahdi, S.; Van Keymeulen, A.; Brown, D.; Moers, V.; Lemaire, S.; De Clercq, S.; Minguijon, E.; Balsat, C.; Sokolow, Y.; Dubois, C.; De Cock, F.; Scozzaro, S.; Sopena, F.; Lanas, A.; D'Haene, N.; Salmon, I.; Marine, J. C.; Voet, T.;  Sotiropoulou, P. A.; Blanpain, C., Identification of the tumour transition states occurring during EMT. Nature 2018, 556 (7702), 463-468.

  1. Ruscetti, M.; Dadashian, E. L.; Guo, W.; Quach, B.; Mulholland, D. J.; Park, J. W.; Tran, L. M.; Kobayashi, N.; Bianchi-Frias, D.; Xing, Y.; Nelson, P. S.; Wu, H., HDAC inhibition impedes epithelial-mesenchymal plasticity and suppresses metastatic, castration-resistant prostate cancer. Oncogene 2016, 35 (29), 3781-95.

   38. Kroger, C.; Afeyan, A.; Mraz, J.; Eaton, E. N.; Reinhardt, F.;    Khodor, Y. L.;  Thiru, P.; Bierie, B.; Ye, X.; Burge, C. B.; Weinberg, R. A., Acquisition of a hybrid E/M state is essential for tumorigenicity of basal breast cancer cells. Proc Natl Acad Sci U S A. 2019, 116 (15), 7353-7362.

Reviewer 2 Report

In general I found this to be a very useful review of multiple aspects of the recent emerging interest in phenotypes that are hybrids with both epithelial and mesenchymal properties. I only have  a few suggestions for improving this manuscript. One of these suggestions is in the area of English editing. Some paragraphs, for example the first one on page 2, clearly have English usage problems. Others are more or less fine.

I think the discussion of the recent reference from the Weinberg group (ref 110) misses an essential point in this paper. It has not been clear whether the apparent increase in metastatic capability of hybrid EMT cells is due to their ability to differentiate into a mixture of E and M cells which then mutually support each other in populating the metastatic site or whether the plasticity inherent in the hybrid itself is the key feature; this paper clearly supports the latter, albeit in only one case, and should be explained more clearly.

Next, the issue of the best surface marker for hybrid cells in the breast cancer context is more complicated than is implied here - for example, several papers (Goldman et al - the cited ref 95, Grosse-Wilde, Anne, et al. "Stemness of the hybrid epithelial/mesenchymal state in breast cancer and its association with poor survival." PloS one 10.5 (2015), Jolly, Mohit Kumar, et al. "Inflammatory breast cancer: a model for investigating cluster-based dissemination." NPJ Breast Cancer 3.1 (2017): 1-8.) argue that the correct marker should be Cd24+/CD44+; for a different view see  Bierie, Brian, et al. "Integrin-β4 identifies cancer stem cell-enriched populations of partially mesenchymal carcinoma cells." Proceedings of the National Academy of Sciences114.12 (2017): E2337-E2346. This should also be discussed in more detail.

It is still controversial to what extent EMT is necessary for metastasis. The authors do cite one of the relevant papers in this regard (ref 67) but perhaps should also discuss a very recent addition to this literature, Lourenco, Ana Rita, et al. "Differential contributions of pre-and post-EMT tumor cells in breast cancer metastasis." Cancer research 80.2 (2020): 163-169. In my opinion there is a very important issue here as to whether we know enough about the markers of partial EMT to be able to do the type of lineage tracing reported here and definitively decide if any cells should be labeled as hybrids; since this is a crucial issue for the field, this should also be discussed in the context of this recent study.

Author Response

Comment 1:

One of these suggestions is in the area of English editing. Some paragraphs, for example the first one on page 2, clearly have English usage problems. Others are more or less fine.

Authors’ reply:

Thank you for the suggestion. The manuscript has been sent for English editing. The certificate of the English editing is attached.

Comment 2:

I think the discussion of the recent reference from the Weinberg group (ref 110) misses an essential point in this paper. It has not been clear whether the apparent increase in metastatic capability of hybrid EMT cells is due to their ability to differentiate into a mixture of E and M cells which then mutually support each other in populating the metastatic site or whether the plasticity inherent in the hybrid itself is the key feature; this paper clearly supports the latter, albeit in only one case, and should be explained more clearly.

Authors’ reply:

Thank you for the excellent suggestion, and we have addressed this point in more detail in line 1296-1299 of the revised manuscript (line 305-308 without track changes). The description is as below:

This study highlights an important issue: the coexistence of epithelial and mesenchymal traits within individual cells rather than the dynamic changes between epithelial and mesenchymal states is required for tumor initiation. This result suggests that the plasticity inherent in the hybrid cells themselves is the key feature of metastasis.

Comment 3:

Next, the issue of the best surface marker for hybrid cells in the breast cancer context is more complicated than is implied here - for example, several papers (Goldman et al - the cited ref 95, Grosse-Wilde, Anne, et al. "Stemness of the hybrid epithelial/mesenchymal state in breast cancer and its association with poor survival." PloS one 10.5 (2015), Jolly, Mohit Kumar, et al. "Inflammatory breast cancer: a model for investigating cluster-based dissemination." NPJ Breast Cancer 3.1 (2017): 1-8.) argue that the correct marker should be Cd24+/CD44+; for a different view see Bierie, Brian, et al. "Integrin-β4 identifies cancer stem cell-enriched populations of partially mesenchymal carcinoma cells." Proceedings of the National Academy of Sciences114.12 (2017): E2337-E2346. This should also be discussed in more detail.

Authors’ reply:

Thank you for the suggestion. We have addressed and discussed more detail description in line 1299-1305 (line 308-314 without track changes). The description is as below:

Other studies used different markers to define hybrid E/M, such as co-expression of CD24 and CD44 (CD24hi CD44hi) in breast cancer cells. The result also showed that the hybrid E/M signature is associated with the stem-like features of breast cancer cells and the poorest outcome in luminal and basal breast cancer patients [126,127]. Although the findings are consistent with regard to the impact of the hybrid E/M state on cancer stemness, it is important to carefully interpret the results using different markers to define the hybrid E/M state.

Comment 4:

It is still controversial to what extent EMT is necessary for metastasis. The authors do cite one of the relevant papers in this regard (ref 67) but perhaps should also discuss a very recent addition to this literature, Lourenco, Ana Rita, et al. "Differential contributions of pre-and post-EMT tumor cells in breast cancer metastasis." Cancer research 80.2 (2020): 163-169.

Authors’ reply:

We have included this paper as the reviewer suggested and discussed this point in the article as lines 395-620 (line 140-145 without track changes). The description is as below:

These controversial results raise the question of whether EMT is essential for the establishment of metastasis. By using fluorescence-based lineage tracing technology, Lourenco et al. showed that in a breast cancer model, pre-EMT cells play a predominant role in metastasis, whereas post-EMT cells support tumor invasion and angiogenesis. Interestingly, post-EMT cells are not permanently committed to the mesenchymal phenotype and are capable of undergoing MET to the epithelial phenotype, suggesting the presence of epithelial–mesenchymal plasticity in these cells [69].

Comment 5:

In my opinion there is a very important issue here as to whether we know enough about the markers of partial EMT to be able to do the type of lineage tracing reported here and definitively decide if any cells should be labeled as hybrids; since this is a crucial issue for the field, this should also be discussed in the context of this recent study.

Authors’ reply:

Regard to the suggestion, and we added a paragraph and the discussion section of the revised manuscript. The description is as below:

In line 351-357 (line 96-102 without track changes) of the revised manuscript:

Compared with the clearly defined epithelial and mesenchymal states, the hybrid E/M seems to lack universal markers to indicate the state. Most studies applied co-expression of epithelial and mesenchymal markers to indicate the hybrid E/M state [9,27,28,31]. Regarding the hybrid state-specific markers, a recent study shows that expression of CD 51, CD61, and CD106 indicates different hybrid E/M states [28]. Expression of CD104 was also noted in the hybrid E/M populations [38]. The transcriptional factor ΔNP63 engenders the entrance of an early hybrid E/M state [28]. The reported hybrid E/M markers/transcriptional factors are summarized in Figure 1.

In addition, we also mentioned this point in the Conclusion of the manuscript , line 1321-1327 of the revised manuscript (line 302-325 without track changes):

Specifically, previous studies mostly applied co-expression of epithelial and mesenchymal markers to define the hybrid E/M state. Nevertheless, co-expression of these markers may not be able to reflect the highly plastic characteristics of the hybrid E/M state, and defined markers for the hybrid E/M state will be important for both experimental and clinical characterization. Application of single-cell analysis techniques will be extremely helpful in solving this issue.

Reference:

  1. George, J. T.; Jolly, M. K.; Xu, S.;  Somarelli, J. A.; Levine, H., Survival outcomes in cancer patients predicted by a partial EMT gene expression scoring metric. Cancer Res. 2017, 77 (22), 6415-6428.

  1. Mizukoshi, K.; Okazawa, Y.; Haeno, H.; Koyama, Y.; Sulidan, K.; Komiyama, H.; Saeki, H.; Ohtsuji, N.; Ito, Y.; Kojima, Y.; Goto, M.; Habu, S.; Hino, O.; Sakamoto, K.; Orimo, A., Metastatic seeding of human colon cancer cell clusters expressing the hybrid epithelial/mesenchymal state. Int J Cancer. 2019.

  1. Pastushenko, I.; Brisebarre, A.; Sifrim, A.; Fioramonti, M.; Revenco, T.; Boumahdi, S.; Van Keymeulen, A.; Brown, D.; Moers, V.; Lemaire, S.; De Clercq, S.; Minguijon, E.; Balsat, C.; Sokolow, Y.; Dubois, C.; De Cock, F.; Scozzaro, S.; Sopena, F.; Lanas, A.; D'Haene, N.; Salmon, I.; Marine, J. C.; Voet, T.;  Sotiropoulou, P. A.; Blanpain, C., Identification of the tumour transition states occurring during EMT. Nature 2018, 556 (7702), 463-468.

  1. Ruscetti, M.; Dadashian, E. L.; Guo, W.; Quach, B.; Mulholland, D. J.; Park, J. W.; Tran, L. M.; Kobayashi, N.; Bianchi-Frias, D.; Xing, Y.; Nelson, P. S.; Wu, H., HDAC inhibition impedes epithelial-mesenchymal plasticity and suppresses metastatic, castration-resistant prostate cancer. Oncogene 2016, 35 (29), 3781-95.

38 Kroger, C.; Afeyan, A.; Mraz, J.; Eaton, E. N.; Reinhardt, F.;  Khodor, Y. L.;  Thiru, P.; Bierie, B.; Ye, X.; Burge, C. B.; Weinberg, R. A., Acquisition of a hybrid E/M state is essential for tumorigenicity of basal breast cancer cells. Proc Natl Acad Sci U S A. 2019, 116 (15), 7353-7362.

  1. Lourenco, A. R.; Ban, Y.;  Crowley, M. J.;  Lee, S. B.;  Ramchandani, D.;  Du, W.;  Elemento, O.;  George, J. T.;  Jolly, M. K.;  Levine, H.;  Sheng, J.;  Wong, S. T.;  Altorki, N. K.; Gao, D., Differential contributions of pre- and post-emt tumor cells in breast cancer metastasis. Cancer Res. 2020, 80 (2), 163-169.

  1. Grosse-Wilde, A.; Fouquier d'Herouel, A.;  McIntosh, E.;  Ertaylan, G.;  Skupin, A.;  Kuestner, R. E.;  del Sol, A.;  Walters, K. A.; Huang, S., Stemness of the hybrid Epithelial/Mesenchymal State in Breast Cancer and Its Association with Poor Survival. PLoS One 2015, 10 (5), e0126522.

  1. Jolly, M. K.; Boareto, M.;  Debeb, B. G.;  Aceto, N.;  Farach-Carson, M. C.;  Woodward, W. A.; Levine, H., Inflammatory breast cancer: a model for investigating cluster-based dissemination. NPJ Breast Cancer 2017, 3, 21.
